# Association between Mother’s Education and Infant and Young Child Feeding Practices in South Asia

**DOI:** 10.3390/nu14071514

**Published:** 2022-04-05

**Authors:** Md. Tariqujjaman, Md. Mehedi Hasan, Mustafa Mahfuz, Muttaquina Hossain, Tahmeed Ahmed

**Affiliations:** 1Nutrition and Clinical Services Division, International Centre for Diarrhoeal Disease Research, Bangladesh (icddr,b), Dhaka 1212, Bangladesh; mustafa@icddrb.org (M.M.); muttaquina@icddrb.org (M.H.); tahmeed@icddrb.org (T.A.); 2Institute for Social Science Research, The University of Queensland, Indooroopilly, QLD 4068, Australia; m.m.hasan@uq.net.au; 3Australian Research Council Centre of Excellence for Children and Families over the Life Course, The University of Queensland, Indooroopilly, QLD 4068, Australia

**Keywords:** infant and young child feeding, mother’s education, South Asia

## Abstract

The association between mother’s education and the World Health Organization’s (WHO’s) eight Infant and Young Child Feeding (IYCF) core indicators has yet to be explored in South Asia (SA). This study aimed to explore the association between mother’s education and the WHO’s eight IYCF core indicators in SA. We analyzed data from the most recent nationally representative Demographic and Health Surveys of six South Asian Countries (SACs)—Afghanistan, Bangladesh, India, Maldives, Nepal, and Pakistan. We found significantly higher odds (adjusted odds ratio, AOR, 1.13 to 1.47) among mothers who completed secondary or higher education than among mothers with education levels below secondary for the following seven IYCF indicators: early initiation of breastfeeding (EIBF), exclusive breastfeeding under 6 months (EBF), the introduction of solid, semisolid or soft foods (ISSSF), minimum dietary diversity (MDD), minimum meal frequency (MMF), minimum acceptable diet (MAD), and consumption of iron-rich or iron-fortified foods (CIRF); the exception was for the indicator of continued breastfeeding at one year. Country-specific analyses revealed significantly higher odds in EIBF (AOR 1.14; 95% CI: 1.11, 1.18) and EBF (AOR 1.27; 95% CI: 1.19, 1.34) among mothers with secondary or higher education levels in India. In contrast, the odds were lower for EIBF in Bangladesh and for EBF in Pakistan among mothers with secondary or higher education levels. For country-specific analyses for complementary feeding indicators such as ISSSF, MDD, MMF, MAD, and CIRF, significantly higher odds (AOR, 1.15 to 2.34) were also observed among mothers with secondary or higher education levels. These findings demonstrate a strong positive association between mother’s education and IYCF indicators. Strengthening national policies to educate women at least to the secondary level in SACs might be a cost-effective intervention for improving IYCF practices.

## 1. Introduction

Globally, inappropriate and suboptimal feeding practices during the first year of life are responsible for two-thirds of child deaths [1]. In low- and middle-income countries (LMICs), only one-half of infants are put to the breast within one hour of birth [2]. Below the optimal level, only 37% of children 0–5 months of age in LMICs are exclusively breastfed [3]. According to the 2010 Global Burden of Disease study, suboptimal breastfeeding was one of the three main causes of disease across much of the Sub-Saharan African region [4]. In South Asia (SA), poor nutritional status and suboptimal and inappropriate IYCF practices prevail [5]. Globally, about 50% of undernourished children reside in SA [6]. The early initiation of breastfeeding and exclusive breastfeeding rates are also low in this region, at 39% and 46%, respectively [7]. Although half of the children start timely complementary feeding, the rates of dietary diversity and meal frequency remain low in SA [8]. 

The World Health Organization (WHO) has recommended initiating breastfeeding within one hour of birth, exclusive breastfeeding for up to six months with continued breastfeeding at one year, and the timely introduction of complementary feeding with feedings including the recommended number of diversified foods [9]. Early initiation of breastfeeding and exclusive breastfeeding practices can reduce neonatal mortality by 22% and mortality in children under two years of age by 13.8%, especially in LMICs [10,11]. To ensure optimum growth and development of a child, the timely introduction of complementary feeding with sufficient quantities and qualities of complementary foods is compulsory [12,13,14]. Appropriate feeding during the first two years of life reduces vulnerability to illnesses such as acute respiratory infections, diarrheal diseases, undernutrition and also reduces childhood mortality to approximately 6% [15,16,17].

Various studies have been conducted in SA to explore the associated factors of IYCF practices [18,19,20,21,22] and have identified a range of detrimental factors, including but not limited to the place of residence, household wealth status, household income, cultural beliefs, parental education, mother’s age, mother’s occupation, antenatal and postnatal care, child’s age, child’s sex, modes of delivery, place of delivery, and women’s and mother’s exposure to mass media [18,19,20,21,22]. Among all these factors, the role of mother’s education is beneficial to health and nutritional well-being [23,24,25,26,27,28]. Some single-country-specific studies also found a positive association between mother’s education and IYCF practices [5,29,30]. However, the data that were used in these studies were neither up-to-date nor did they cover all South Asian Countries (SACs). The assessment of all the WHO’s IYCF core indicators is also scarce in SA. Furthermore, there is a lack of pooled evidence regarding the association between mother’s education and IYCF practices. The pooled estimates at the regional level are particularly important for cross-regional comparison and for designing and implementing interventions at the regional level. Therefore, there is a need for strong evidence both for pooled and country-specific data regarding the association between mother’s education and the WHO’s IYCF core indicators.

In our study, we hypothesized that there is a positive association between mother’s education and the WHO’s IYCF core indicators. Mother’s education is positively associated with the nutritional status of children [23]. Also, mother’s education is strongly associated with access to healthcare facilities or skilled birth attendants for childbirth [24], childhood immunization through vaccination [25], and hygiene practices [26]. By its profound impact on healthcare-seeking behavior, mother’s education has helped to reduce childhood undernutrition and mortality [27,28]. Given the importance of mother-child interactions and the role of mothers in childcare and health-related practices, we considered mother’s education as the main exposure variable. However, evidence-based inference on this hypothesis remains unseen in SA. Therefore, in this study, we aimed to explore the association between mother’s education and all of the WHO’s IYCF core indicators in SACs. The findings generated from this study might be directives for policymakers in SACs to design interventions for improving IYCF practices.

## 2. Materials and Methods

### 2.1. Data Source and Study Design

This study analyzed the most recent nationally representative data of six SACs—Afghanistan (2015), Bangladesh (2017–2018), India (2016), Maldives (2017), Nepal (2016), and Pakistan (2018). Though Bhutan and Sri Lanka are on the list of SACs, we excluded these two countries because in Bhutan the DHS program is not functioning, and the latest Sri Lankan DHS data are restricted. The DHS follows a similar methodology to collect cross-sectional data by applying a multistage cluster sampling technique. The use of the unique methodology of the DHS allows the cross-country comparison of estimates. The DHS compiles information on a variety of health-related indicators and their sociodemographic factors, including the socioeconomic status of the household, women’s empowerment, and healthcare-seeking behaviors. The interviews are conducted only where the respondent gives voluntary informed consent. The DHS data are publicly available from the DHS program website (https://dhsprogram.com/, accessed on 3 April 2020) [31]. 

### 2.2. Participants 

The participants in this study were the youngest children, aged 0–23 months, and their mothers, aged 15–49 years. The DHS usually collects data on IYCF practices for the most recent birth child of the mother. Therefore, we included mothers and their most recent birth child of 0–23 months of age as our study participants.

### 2.3. Outcome Variables

The outcome variables of this study were the eight WHO IYCF core indicators: early initiation of breastfeeding (EIBF), exclusive breastfeeding under 6 months (EBF), continuing breastfeeding at 1 year (CBF), the introduction of solid, semi-solid or soft foods (ISSSF), minimum dietary diversity (MDD), minimum meal frequency (MMF), minimum acceptable diet (MAD), and consumption of iron-rich or iron-fortified foods (CIRF). The outcome variables were dichotomous (yes, no). We used the standard definitions of the WHO for constructing the IYCF core indicators [9].

### 2.4. Exposure Variables

The main exposure variable of this study was the level of mother’s education. Based on the percentage distribution among the different levels of mother’s education status (no education, primary, secondary, and higher education), we categorized mother’s education into two levels: below secondary (no education or primary-level education) and secondary or above (secondary education or higher). We also considered the years of mother’s schooling as a continuous variable for sensitivity analysis. In most cases, the mother becomes the primary caregiver of a child and mother’s knowledge of feeding practices directly reflects the nutrition intake of the child. Mother’s education also has multiple benefits for improving the nutritional status of children as well. Given the importance of mother’s education, we selected it as our main exposure variable for establishing the linkage between mother’s education and IYCF practices.

### 2.5. Independent Variables 

The independent variables of this study were mother’s age (categorized as 15–24 years, 25–34 years, and ≥35 years), age of the children (continuous), sex of the children (male and female), type of place of residence (urban and rural), and wealth index (poorest, poorer, middle, richer, and richest). We included some selective independent variables in our study. The selection of independent variables was made in terms of their commonness, significance, and existence in the existing literature or dataset [32,33,34,35,36]. Therefore, the association between mother’s education and IYCF practices was explored by controlling the effect of these selected independent variables. We also included the study years (2015, 2016, 2017 and 2018) in which the studies were conducted to control for variations due to different survey periods in the pooled estimates.

### 2.6. Statistical Analyses

All the data analyses were performed by the statistical software package Stata, version 15.0 SE (College Station, TX, USA). The weighted estimates of sociodemographic and IYCF indicators were performed for both pooled and country-specific analyses. We adjusted the effect of complex survey design, including for country-specific sampling weights and clusters, while estimating the percentages. For pooled datasets, we denormalized the sampling weight and created a new population-level weight by dividing the sampling weight by the denormalized weight. We also constructed a unique cluster variable by combining country and cluster numbers. The population-level weight and unique cluster were used to calculate the pooled estimates. The population-level weight was calculated to avoid the effect of countries with a large population (such as India) balancing countries with a smaller population (such as the Maldives). 

We performed generalized estimating equation (GEE) models for pooled and country-specific data to explore the association between mother’s education and IYCF indicators at both the regional and country levels, respectively. In the multiple GEE models, we adjusted the models by mother’s age, child’s age, child’s sex, place of residence, and wealth index. We used the log-binomial model with logit link in the GEE models. When the log-binomial model was unable to achieve convergence, we used Poisson models with log links. The strength of association was measured in adjusted odds ratios (AORs) with respective 95% confidence intervals (CIs). We also performed a sensitivity analysis that considered mother’s education as a continuous variable (years of schooling) to explore the pattern of association. A *p*-value of <0.05 was considered to determine the statistical significance of all the two-sided tests performed. We checked the collinearity among the independent variables and found the variance inflation factor <1.50, indicating negligible collinearity among the independent variables. We performed a pooled analysis since a pooled estimate is crucial to obtaining a firm conclusion from the results or data of different studies when the same methodology and analysis techniques are applied. The complete observations of the outcomes, exposure, and independent variables were included in the regression models. The extraction of the total sample size used in the regression models is presented in Figure 1.

## 3. Results

### 3.1. Sociodemographic Characteristics of the Study Participants

The analysis included 120,830 mother–child (youngest) dyads. Overall, the mean age of the children was 11 months, with half of the children being female. Nearly half (47.3%) of the mothers of the index children were aged between 25 and 34 years, while a little more than half of the mothers (54%) had obtained secondary or higher education levels. The majority of the children were from rural areas (72%) and a little less than a quarter (24%) belonged to the poorest households. These estimates varied across SACs, with the child’s rural residence highest in the Maldives (90%), with the secondary or higher education of the mother lowest in Afghanistan (11%), and with belonging to the poorest household highest in the Maldives (27%). See Appendix A for details.

### 3.2. Prevalence of IYCF Practices 

In SACs, overall, about 45% of mothers practiced EIBF, whereas the prevalence of EBF and CBF were 54% and 83%, respectively. In the case of complementary feeding, overall, the ISSSF, MDD, MMF, MAD, and CIRF practices were 50%, 22%, 40%, 13%, and 21%, respectively. The lowest prevalence of EIBF was noticed in Pakistan (21%), of EBF in Afghanistan (42%), and of MAD in India (13%) (Appendix A). 

### 3.3. Association between Mother’s Education and IYCF Practices 

In the bivariate analysis of the pooled data, we found significantly higher IYCF (except for CBF) practices among higher educated mothers compared with mothers with no formal or below secondary-level education. In the country-specific estimates, similar higher IYCF practices were observed among mothers with secondary or higher levels of education. In contrast, in Bangladesh, significantly higher EIBF practices, and in Pakistan significantly higher EBF and CBF, were observed among women with no formal or below secondary-level education (Appendix A). 

In the multiple GEE models of the pooled data, we found significantly higher odds (AOR range, 1.13 to 1.47) of IYCF practices among mothers with secondary or higher education levels than mothers with below secondary-level education. These findings across SACs show that the odds were highest among mothers with secondary or higher education levels compared with mothers with below secondary education levels for EIBF (AOR 1.14; 95% CI: 1.11, 1.18) in India and for EBF (AOR 1.27; 95% CI: 1.19, 1.34) in India also. In contrast, mothers with secondary or higher education had the lowest odds compared with mothers with below secondary-level education for EIBF in Bangladesh (AOR 0.84; 95% CI: 0.72, 0.99) and for EBF in Pakistan (AOR 0.64; 95% CI: 0.48, 0.85). In the case of complementary feeding (ISSSF, MMD, MMF, MAD, and CIRF) in the pooled data, significantly higher odds (AOR range: 1.17 to 1.47) were found among mothers with secondary or higher education levels. These findings for country-specific analyses show the odds were highest among mothers with secondary or higher education compared with mothers with below secondary-level education for ISSSF (AOR 1.55; 95% CI: 1.03, 2.32) in Pakistan, and for MDD (AOR 2.25; 95% CI: 1.72, 2.94), MMF (AOR 1.61; 95% CI: 1.24, 2.08), MAD (AOR 2.34; 95% CI: 1.76, 3.11), and CIRF (AOR 1.47; 95% CI: 1.14, 1.91) in Nepal (Figure 2). 

### 3.4. Sensitivity Analyses

We performed sensitivity analyses and considered our exposure variable as continuous (mother’s completed years of schooling). Sensitivity analyses included the multiple GEE modeling of mother’s completed years of schooling and all IYCF indicators, adjusting the similar independent variables as performed in the main regression analysis. The objective of the sensitivity analyses was to see how IYCF practices change (either increase or decrease) with each additional year of mother’s schooling. We found that all IYCF practices increased with a one-year increase in mother’s years of completed schooling. Similar results were observed in the country-specific analyses. In contrast, we found significantly lower odds for mother’s education with EIBF in Bangladesh and for mother’s education with EBF in Pakistan. The regression findings for sensitivity analyses (Appendix A) were similar to our main regression analyses findings. 

## 4. Discussion

Ensuring the recommended IYCF practices are present to start a healthy life is the right of every child. To our knowledge, this is the first study to explore the association between mother’s education and the WHO’s eight IYCF core indicators in SACs by using both pooled and country-specific data from the latest DHS. The findings of our pooled results indicated that mothers with secondary or higher education levels had significantly higher odds of practicing all of the WHO’s IYCF core indicators except for continued breastfeeding at one year. In the country-specific analyses, we also found a similar, significant positive association between mother’s education and IYCF indicators, but we found a significant negative association between EIBF and mother’s education in Bangladesh and between EBF and mother’s education in Pakistan. 

Our findings on the higher rate and likelihood of IYCF practices among mothers with secondary or higher levels of education coincide with previous studies. In particular, children of mothers with secondary or higher levels of education reported a higher likelihood of EIBF in Ghana [32], Ethiopia [33,34], Tanzania [35], and China [36]; of EBF in the Kingdom of Saudi Arabia [37], Indonesia [38], and Sub-Saharan Africa [39]; of introducing complementary feeding in Northwest [40] and Southwest Ethiopia [41]; and of MDD, MMF, MAD, and CIRF in Ethiopia, Zambia, and the Sub-Saharan African region [42,43,44,45,46]. The positive association between higher maternal education and improved IYCF practices was well-expected. Educated mothers are likely to be more exposed to mass media, to have adequate knowledge about health, nutrition, and feeding practices, and to understand the importance of feeding diversified foods to their offspring [42]. Educated mothers are also likely to have more opportunities to earn income and make decisions regarding the spending money of their health and their children’s health care services [47]. There is evidence that earning income and having autonomy in making decisions are associated with optimal IYCF practices [48]. Educated mothers might have the ability to adopt social and behavioral changes and to practice them in real life [49]. Interventions in social and behavioral change communication among mothers have proven impact on increased IYCF practices, particularly on MMD, MMF, and MAD [50]. These strong positive associations between mother’s education and IYCF practices suggest that policymakers of SACs should increase and sustain maternal education at least through the secondary level by implementing appropriate interventions. 

In contrary to the positive association between mother’s education and IYCF practices, we found negative associations between high maternal education and low IYCF practices, particularly practices related to breastfeeding (such as EIBF in Bangladesh and EBF in Pakistan). A systematic review conducted in developed countries found that women with the highest level of education had a 2.28 times higher probability of not initiating breastfeeding within one hour of birth [51], which supports our finding. Another study conducted on 81 LMICs found a negative association between mothers’ education and EIBF [52]. A significant negative association between mother’s education and EBF practices was also observed in China, Saudi Arabia, the United Arab Emirates, and Lebanon [36,53,54,55,56]. A previous study in Bangladesh found that educated mothers were more prone to undergo caesarean section (C-section) delivery [57]. The C-section delivery rate is the highest (59%) in Bangladesh among SACs and South-East Asian countries [58]. Evidence has suggested that C-section delivery is one of the major risk factors for not initiating breastfeeding early after birth [22,59,60]. This might be the reason why mother’s education is negatively associated with EIBF in Bangladesh. To improve EIBF in Bangladesh, lowering the rate of C-section delivery in private facilities, and the counselling of mothers by doctors during antenatal visits highlighting the negative effects of C-section delivery might be effective. Also, skin-to-skin contact immediately after C-section delivery was found to be a significant predictor for improving EIBF practices in LMICs [61]. The reason for the negative association between mother’s education and EBF in Pakistan is that highly educated women have more opportunities to engage in formal employment, and there was evidence that most of the highly educated mothers were employed [62]. To improve EBF in SACs, multiple visits at the beginning of and during pregnancy, interpersonal education and counseling, community mobilization, and mass media exposure were found to be effective [63]. Also, to improve EBF, emphasis should be given to educating women in SACs through the strengthening of policies to prevent early marriage, through interventions for building awareness of female education, and through policies for ensuring education for females. Strengthening nutritional education for mothers, caregivers, family members, and relatives, and building community awareness might be helpful for the timely introduction of complementary feeding [64]. In addition to that, we suggest the policymakers in SACs ensure education for women through proper policies which ultimately help women to improve appropriate knowledge of the timely initiation of complementary feeding. 

Our study has several strengths. The pooled estimate of SACs provided firm evidence of the association between mothers’ education and IYCF practices. The country-specific analyses will help policymakers to set appropriate interventions for women’s education which help to achieve optimal IYCF practices. The positive association between mother’s education and the WHO’s core indicators has generated new evidence for the South Asian context. Our study has some limitations, too. First of all, we cannot make causal inferences due to the cross-sectional nature of the data. However, to explore the association between outcome and exposure, cross-sectional studies are well accepted. Second, the measurements of breastfeeding and the child’s diet-related responses were based on mother’s 24-hour recall reports, which may include mother’s subjective response and recall bias. However, in the case of calculating breastfeeding and the child’s diet-related IYCF indicators, without mother’s 24-hour recall response, there is a lack of an alternative method. Third, for calculating the IYCF indicators, we included only children under two years of age and their mothers; also, for calculating some indicators, such as EBF, CIBF, and ISSSF, we used sub-samples of the child’s specific age, which reduced the estimated sample size. 

## 5. Conclusions

In summary, IYCF practices in SA are still low. Overall, our findings indicated that mother’s education has a significantly and consistently positive association with IYCF practices in SA. The strong positive association between IYCF practices and mother’s education indicates mother’s education has a significant role in improving IYCF practices. To improve IYCF practices in SACs, one of the major cost-effective interventions might be educating females at least through the secondary level. The national-level policies in each SAC must be prioritized to ensure at least secondary-level education for women and mothers to ensure optimal IYCF practices. 

## Figures and Tables

**Figure 1 nutrients-14-01514-f001:**
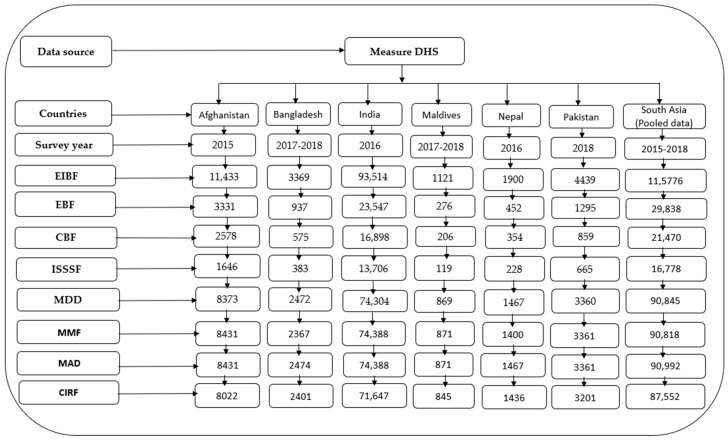
Sample flowchart for multiple regression models. DHS = Demographic and Health Surveys, EIBF = early initiation of breastfeeding, EBF = exclusive breastfeeding, CBF = continued breastfeeding at 1 year, ISSSF = introduction of solid, semi-solid and soft foods, MDD = minimum dietary diversity, MMF = minimum meal frequency, MAD = minimum acceptable diet, and CIRF = consumption of iron-rich or iron-fortified foods.

**Figure 2 nutrients-14-01514-f002:**
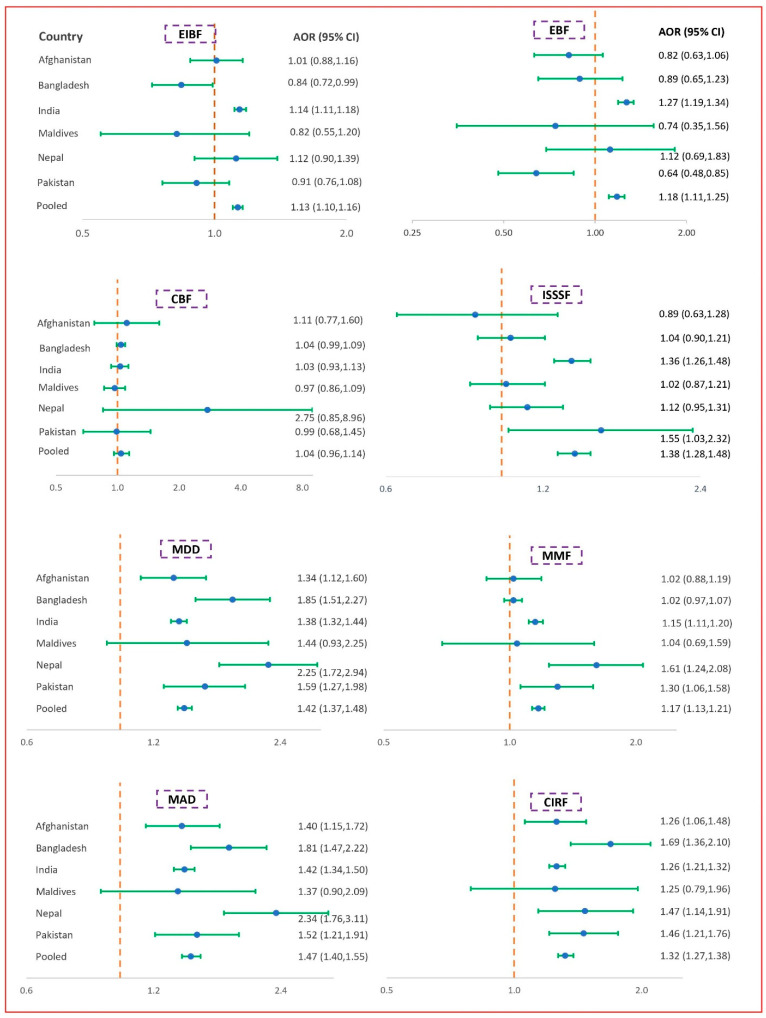
Association between mother’s education and IYCF indicators. The *x*-axis of the graph represents AOR (adjusted odds ratio) and the *y*-axis represents different pooled and SACs data; CI = confidence interval; adjusted odds ratios, 95% confidence intervals, and *p*-values were obtained from the log-binomial regression model using the generalized estimating equation. Models adjusted for child’s age, child’s sex, maternal age, place of residence, and wealth index. For pooled data, in addition to these factors, we adjusted the model by survey year. EIBF = early initiation of breastfeeding, EBF = exclusive breastfeeding, CBF = continued breastfeeding at 1 year, ISSSF = introduction of solid, semi-solid and soft foods, MDD = minimum dietary diversity, MMF = minimum meal frequency, MAD = minimum acceptable diet, and CIRF = consumption of iron-rich or iron-fortified foods.

## Data Availability

The data of this study are publicly available. Data can be downloaded at https://dhsprogram.com/data/available-datasets.cfm, accessed on 3 April 2020.

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
