# Peer review of "Association between Mother’s Education and Infant and Young Child Feeding Practices in South Asia"

_nutrients, 2022, doi:10.3390/nu14071514_

Round 1

Reviewer 1 Report

A nice article. Although it does not report a novel finding, it strengthens existing knowledge, which is of public health importance. Please see the attachment for my comments for your consideration. Please take the time to critically synthesize the message, avoid unnecessary repetitions, and ensure a streamlined flow. 

Author Response

Comments and Suggestions for Authors

A nice article. Although it does not report a novel finding, it strengthens existing knowledge, which is of public health importance. Please see the attachment for my comments for your consideration. Please take the time to critically synthesize the message, avoid unnecessary repetitions, and ensure a streamlined flow. 

Review: Association between Mother’s Education and Infant and Young Child Feeding Practices in South Asia

Global comments:                                                  

  • Revise the manuscript to make it concise and streamlined.

o For example, in the second paragraph in the introduction, references 10, 11, 16, and 17 refer to a similar message. Therefore, you could rephrase the paragraph, shorten it, and reference it appropriately. Please go through the whole manuscript and see if you see this kind of unnecessary repetitions and revise accordingly.

Response: Thank you so much for your comments. According to your suggestion, we have revised the whole manuscript where we found duplication. The revised second para is now as follows:

“Early initiation and exclusive breastfeeding practice can reduce respectively 22% of neonatal mortality and 13.8% of under two deaths, especially in LMICs [10, 11]. Appropriate feeding during the first two years of life reduces the vulnerability of illnesses such as acute respiratory infections, diarrhoeal diseases, undernutrition, and approximately 6% childhood mortality [15, 16, 17].”

o The ideas and sentences are disjointed in the discussion section, not telling a streamlined, coherent story. There is significant repetition and a lack of critical analysis (mainly descriptive).

Response: Thank you. According to your suggestion, we have tried to streamline the discussion section tried to avoid repetition, and critically analyzed the results.  

  • The phrase “illiterate mothers” seems a bit harsh; could you use “mothers with no formal education”?

Response: Thanks a lot for your valuable suggestion. According to your suggestion, we changed the term “illiterate mothers” to “mothers with no formal education” throughout the manuscript in the revised version.

  • Should the focus be on educating women only or couples? Is there evidence that men’s education does not influence feeding practices? If it does, you could state that this was a confounding factor. E.g: doi: 10.1007/s13312-014-0471-3. PMID: 25129001.

Response: Thanks for pointing this. We also believe fathers' education also has a positive effect on IYCF practices. In the initial stage, we included both father's and mother’s education, but we found a higher proportion of missing values in the father’s education, especially in India, which might affect the regressions models. That’s why we excluded fathers’ education from our study.  

Specific comments:

  • Below are suggested changes for your consideration, it is not an exhaustive list.

Line

Comment

12

Please define IYCF as it is the first occurrence

Response: Thanks. We have spelled-out IYCF in the revised version.

35

Change to “low- and middle-“

Response: Changed “low- and- middle- “to “low- and middle-“

39

Delete the word “regions” or change it to “region”.

Response: Deleted

44

Delete the word “in”

Response: Deleted

60 - 61

“household income” repeated, and household wealth mentioned, is this needed?

Response: Deleted household income. We think we can keep household wealth. Because household wealth status is constructed from different indices using principal component analysis and it is a good indicator to display the socioeconomic status of the respondents. 

64, 71

Delete the word “individual”

Response: Deleted

74

Delete “all”

Response: Deleted

77 - 79

Statement referenced [29,30], please4 move it up, earlier in the introduction

Response:

Thank you. We mentioned in this para the rationale and importance of including maternal education as our main exposure variable. Hence we can keep these two references in this para.

257

“ with illiterate” you may want to choose a different word for “with”

Response: Changed

264

Replace “limiting” with “initiation”

Response: changed

267

caesarean

Response: Changed “caesarian” to “caesarean”

272

273

Please rephrase sentence, ..” by counseling to the higher educated mother might be effective”

Response: We have rephrased the sentences as “To improve the EIBF in Bangladesh, lowering the rate of c-section delivery in private facilities, counselling with mothers by doctors during anti-natal visit highlighting the negative effects of c-section delivery might be effective” (page 9; lines: 317-319).

296

“in Pakistan” repeated

Response: Thanks for noting this. Kept only one.

300

Educating women through “proper policies and programs” what does it mean?

Response: changed to asthrough strengthening policies for preventing early marriage, interventions for building awareness for female education and policies for ensuring education for female” (page 9; lines: 327-328).

455

481                      

Please update reference no.31 and no.41. For reference no. 31 you could add the website you cite at no.100 but include the access date  

Response: We updated the references in the revised version.

Reviewer 2 Report

I found very interesting reading the manuscript. The manuscript has been written well. Have just a few minor edits to improve the paper.

Abstract, line 12: WHO, IYCF- spell out when abbreviation is used for the first time, and do the same in the entire paper.

Line 153: generalized estimating equation (GEE)

Line 353: South Asia instead of SA

Suggestion: If possible, run the same model with the father's education.

Thank you.

Author Response

Comments and Suggestions for Authors

I found very interesting reading the manuscript. The manuscript has been written well. Have just a few minor edits to improve the paper.

Abstract, line 12: WHO, IYCF- spell out when abbreviation is used for the first time, and do the same in the entire paper.

Response: Thank you for pointing these. We have spelled it out in the revised version.

Line 153: generalized estimating equation (GEE)

Response: We have incorporated the revised version.

Line 353: South Asia instead of SA

Response: We have incorporated the revised version.

Suggestion: If possible, run the same model with the father's education.

Thank you.

Response: Thanks for your suggestion. In the initial stage, we included father’s education in our analysis. But we found a large number of missing values in the Indian sample for the father’s education variable. As we are analyzing the sub-samples for constructing IYCF indicators, a large number of missing values of father education will also lead to a large number of missing values in the regression analysis. For this reason, we excluded the father’s education from our analyses.